# Efficient Softmax Approximation for Deep Neural Networks with Attention Mechanism

## Abstract

There has been a rapid advance of custom hardware (HW) for accelerating the
inference speed of deep neural networks (DNNs). Previously, the softmax layer
was not a main concern of DNN accelerating HW, because its portion is relatively
small in multi-layer perceptron or convolutional neural networks. However, as the
attention mechanisms are widely used in various modern DNNs, a cost-efficient
implementation of softmax layer is becoming very important. In this paper, we
propose two methods to approximate softmax computation, which are based on
the usage of LookUp Tables (LUTs). The required size of LUT is quite small
(about 700 Bytes) because ranges of numerators and denominators of softmax are
stable if normalization is applied to the input. We have validated the proposed
technique over different AI tasks (object detection, machine translation, speech
recognition, semantic equivalence) and DNN models (DETR, Transformer, BERT)
by a variety of benchmarks (COCO17, WMT14, WMT17, GLUE). We showed
that 8-bit approximation allows to obtain acceptable accuracy loss below 1.0%.

## 1   Introduction

After Vaswani et al. had introduced Transformer model in [27] for machine translation task, the
attention based architecture became popular firstly in Natural Language Processing (NLP) appli-
cations, e.g.: speech recognition [16], [21], [9]; summarization [7]; language understanding [5],
[33], [15], [18]; and video captioning [3]. Recently attention-based models are used in even wider
practical areas including Computer Vision (CV) tasks for object detection [2]; image transformation
[26]; image classification [6]; and even symbolic integration and solving differential equations [14].
Despite the attractiveness of transformer-based models, its direct implementation into the platform
with constrained computational power (e.g., mobile SoC, edge devices) [1] is very challenging due to
big memory footprint and latency.

Therefore, model compression techniques such as quantization, and distillation are needed for those
models. Many approaches have been introduced on quantizing matrix multiplication of transformer
architecture. For example, in [22] it was used second order Hessian information, what allows to
significantly compress the size of the model up to $13\times$ times, while maintaining at most 2.3% of
performance degradation (for the case of ultra-low precision 2-bits quantization). In [35] it was
used a quantization-aware training during the fine-tuning phase of BERT, what allows to compress
BERT model by $4\times$ (with 8-bit quantization) with minimal accuracy loss (less than 1%). In [1], a
machine language translation model was quantized by 8-bit, while maintaining less than 0.5% drop

---

[1]Recently, interest to the computations performed close to the data sources is growing up aiming to soften the
requirements of continuous access to high-speed and high-bandwidth connections. Moreover, often customers
wanted to keep their security and privacy, and thus do not want to expose their data to the external clouds [32],
[19], [36], [11].

in accuracy. Moreover, in [20] it was shown that 8-bit quantized models provide the same or even higher accuracy as the full-precision models. Most of the above methods consider the quantization of matrix multiplications operations only. However, as it is shown in [4], [24] in modern DNNs with attention mechanism (e.g., Transformer, BERT, GPT-x) the softmax function is also used intensively, especially at the longer sequence lengths, so it is necessary to optimize its for performance.

In this paper we propose methods for efficient computation of the softmax layer at the HW accelerator. The method is based on piece-wise-constant approximation and usage of LUTs. To the best of our knowledge, it is the first paper where softmax quantization of the models with attention mechanism is tested and verified on a variety of AI tasks. In Section 2 we show why our research is important and valuable. In Section 3 we consider the drawbacks of existed softmax approximation methods in the perspective of HW accelerator, and summarize the differentiation of our methods from the previous arts. In Section 4 we describe the details of the proposed methods. Section 5 shows the experimental validation over different models and datasets, and Section 6 concludes the paper.

## 2   Background and motivation

Modern GPUs are powerful, but big, expensive, and power-hungry. Therefore, alternative HW accelerators (e.g., NPU) for on-device inference are under active development by different vendors, especially for Federated Learning and Edge computing. However, such devices mostly are focused on the acceleration of matrix multiplication operations, and do not include means to compute complex activation functions. Typically, in such devices the data is sent outside of the accelerator to compute activations on host CPU. For example, according to the guidelines of Coral (TM), a softmax layer of DNN model in Edge TPU have to be run on host CPU [2], what is acceptable for traditional CV tasks (which are typically uni-directional, have minimum dependencies, and softmax layer is located at the end of the computational graph of DNN model), however is very inefficient for NLP tasks (which are typically more complicated with a lot of dependencies and active employment of softmax layer in the middle of DNN model). In opposite to traditional logic-centric approach, some researches are trying to perform computation closer to the memory (so called memory-centric approach). For example in [23], there is shown a DRAM-based AI accelerator. This approach allows significantly speed-up the overall computation process, but for the computation of the activations the data should also be moved to host processor, what is an even bigger issue in the DRAM environment.

The Eq. (1) from [27] describes how attention is computed in the model. This particular form, named "scaled dot-product attention" takes the matrix multiplication product of queries and keys of $\mathbf{R}^{N \times L \times H}$ as input for the softmax layer where $N$ means number of heads, $L$ means sequence length and $H$ means hidden size for the case where batch size equals to 1. In other words, performing $(N \times L \times L)$ softmax operations is required per one attention. Furthermore, encoder in typical transformer consists of six multi-head attentions which means $6 \times (N \times L \times L)$ operation is required for encoder solely. Assuming the number of heads is 8 and sequence length is 128, it already takes $786,432$ operations for softmax of the transformer encoder. This overhead increases as number of heads and sequence length increases which is typical case for high-performing models.

$$Attention(Q, K, V) = Softmax\left(\frac{QK^T}{\sqrt{d_K}}\right)V \tag{1}$$

For example, it requires performing $12 \times (12 \times 128 \times 128) = 2,359,296$ softmax operations for one sample inference for typical BERT configuration [5] over sequence of length 128. In the case when HW accelerator is used for matrix multiplication only, and activation to be computed at the CPU (what is common case for HW accelerators, optimized for CNN-models), the huge amount of data must be moved between CPU and the accelerator. Such data movement negatively impacts on the overall computation time and power consumption, which can be critical for on-device inference. Therefore, HW accelerator must be able to compute softmax layer without CPU involvement.

## 3   Related work and key contributions

The common equation to compute softmax function over the input $x$ is a fraction as shown below:

---

[2]https://coral.ai/docs/reference/edgetpu.learn.backprop.softmax_regression/

$$\sigma(x_i) = \frac{e^{x_i}}{\Sigma e^{x_i}} = \frac{e^{x_i - max(x)}}{\Sigma e^{x_i - max(x)}} \qquad (2)$$

There are different ways to implement it. For example, some approaches straightforwardly compute the numerator and denominator firstly, and then a division operation is performed. In such case the HW accelerator should contain a divider, what requires additional HW costs and can also cause performance degradation, if divider is not fully pipe-lined.

In [25] it is proposed to use basic-split calculation method, which allows to split the exponentiation calculation of the softmax into several specific basics which are implemented by LUT (ROM). It allows to simplify the complexity of hardware and signal propagation delay. However, to recover the whole computed value of exponent some additional multiplications are needed. Moreover, to obtain the final value of softmax the division is still used. In [30] it is proposed to add threshold layers to accelerate the training speed and replace the Euler's base value with a dynamic base value to improve the network accuracy. Such approach allowed to save up to 15% of training model convergence time and also increase by 3 to 5% the average accuracy. But during the computation of softmax the divider is still used. In [17] the combination of LUT and multi-segment polynomial fitting have been used to compute exponential operations of integer and fractional parts in separate. In addition, they adopt radix-4 Booth-Wallace based multiplier for computing the whole value of exponent, and modified shift-compare divider for computation of the final value of softmax. To avoid big area costs for traditional divider, the authors in [8] propose to reduce the operand bit-width, and approximate exponential and division operations with cost-effective addition and bit shifts operations. In their design they have approximated the division operation in Eq. (2) by replacing the denominator with closest $2^b$ value, where $b$ is some integer constant. Then division is implemented just as simple bit shifts operation. In [24] it is proposed to replace the base as $e^x \rightarrow 2^x$, then all computations are more hardware-friendly, however the division operation is still required. Also, to restore accuracy a fine-tuning of the model is needed, what is not applicable for post-training quantization paradigm. Although the methods described above are decreasing the hardware complexity of softmax computation they all still rely on the division operation.

To avoid division operation at all, some other solutions apply the logarithmic transformation to the original softmax function, and thus substitute costly division operation by subtraction of the logarithm. In [34], for example, it is proposed to use a logarithmic operation implemented as a LUT and a subtractor to replace the division operation, what allows to further decrease the complexity of hardware, as well critical path of the whole design. In [10] simplified version of Integral Stochastic Computation is used in order to build FSM-based exponentiation. Division operation is substituted by LUT-based logarithmic operation and subtraction, similarly to [34]. In [31] the authors are further developing the method proposed in [34], by applying mathematical transformations and linear fitting. After optimization, their final design includes only shift operations, leading one detector, and adders. Finally, there are some extreme approximation cases represented in [13] and [28], where logarithmic computation and subtraction are skipped at all.

Despite its attractiveness, logarithmic transformation approach can be used only in the cases when softmax layer is the last layer in DNN and its functionality is simply "scoring" among the candidates for classification tasks. However, if softmax layer is used inside of computational graph of DNN (e.g., DNNs with attention-mechanism) then error caused by quantizations will be accumulated drastically, directly impacting on the final accuracy. For example, in Table 1 there is shown the averaged accuracy drop for DETR models caused by a softmax approximation in uint8 precision by some prior arts. As it can be seen from the Table 1, a straightforward usage of Eq.(2) from [31] causes big accuracy drop, and even after applying some improvements to the original method (shown as case Eq.(2)+), the accuracy drop is still high (2% to 19%). For more details of the prior arts experiments please refer to Appendix A.1. However, if for the same conditions we use the method proposed in Section 4.1, we can see that accuracy drop reduced by $\times 4$ to $\times 20$ times, and it is below $0.5\%$ for plain DETR models (no DC5 dilation at the last stage).

The work presented in this paper has focused on the development of methods for efficient computation of softmax layer during the inference at the edge devices, what usually have limited computational power and suffer from constraints of the bandwidth.

Previous works for HW accelerator of softmax layer are focused on the logic-centric approach and used dedicated hardware for its implementation. In such case the utilization of hardware is low,

Table 1: Averaged accuracy drop by different methods over DETR models (Average Precision), %

| METHOD | DETR (R50) | DETR+DC5(R50) | DETR (R101) | DETR+DC5(R101) |
|---|---|---|---|---|
| EQ.(2) IN [31] | 7.20 | 19.30 | 10.25 | 25.37 |
| EQ.(2)+ IN [31] | 2.50 | 12.93 | 5.38 | 18.85 |
| SECTION 4.1 | **0.33** | 2.92 | **0.22** | 2.73 |

performance can be slower, and no reconfigurability is provided. In our paper we have used an alternative memory-centric approximate computing approach. It keeps accuracy loss small, while allows computing softmax operation with no divider. The size of the required memory (i.e., LUT) is reasonably small and can be reconfigured on demand.

The methods proposed in the paper contribute to building the alternative concept of hardware architecture to accelerate essential operations for AI applications, especially for on-device inference. To summarize, we have three-fold difference from the previous works:

- Applicability of our methods to DNN with **attention mechanism** is experimentally proven over variety of the models for different AI applications. All previous methods were used only for the cases when softmax is the last layer in DNN, and is used for "scoring".

- **No divider** is needed to fully implement the method. Moreover, for 2D LUT method even multiplier is not needed. Thus, hardware overhead is minimal, and is almost free if used in the DRAM-based AI accelerator.

- Our solutions utilize **integer precision**, what makes it compatible with traditional HW accelerators used for matrix multiplication, and simplify the integration of methods into full system (all prior methods are based on a fixed point precision).

## 4 Proposed methods

In this paper we use memory-centric approach to build the accelerator for softmax computation in hardware platform with limited resources. We propose two LUT-based methods for efficient computation, which provide high performance and do not require a divider. The details of the methods are described below and appropriate software models are shown in Appendix A.2.

### 4.1 Normalization of reciprocal exponentiation

In this subsection we consider the method, which is based on the normalization of reciprocal exponentiation, and hereafter we call it REXP for short.

The original reciprocal exponentiation method was proposed in [28], where they used the inverse way of max-normalization and the reciprocal of exponential function as below:

$$\sigma^*(x_i) = \frac{1}{e^{max(x)-x_i}} \tag{3}$$

And thus, the final value of softmax can be obtained by reading from a simple LUT-table. Content of LUT is computed as shown below:

$$LUT_{1/e}[i] = \left\lfloor \frac{1}{e^i} \cdot (2^w - 1) \right\rfloor, \forall i = 0, 1, ..., x_q + 1 \tag{4}$$

where $w$ is a number of bits for quantization, and $x_q = \lceil ln(2^w - 1) \rceil$ is an efficient quantization boundary.

In addition to very low computational complexity, this method has other desired properties [28]:

- it is positive ($\frac{1}{e^x} > 0 \ \forall x \in (-\infty, +\infty)$);

- bounded and stable ($\frac{1}{e^{max(x)-x_i}} \in (0, 1]$);

- and nonlinear ($\frac{1}{e^{\alpha x}} \neq \alpha \frac{1}{e^x}$).

But due to its aggressive approximation nature, it can be applied only to simple CV tasks, and if used for attention-based DNN models causes the explosion of accuracy drop (see Appendix A.1 for details). Thus, in this paper we further develop that method to be applicable for wider class of DNN models.

During our initial investigations, we have noticed that method described in Eq.( 3) is just scaled version of real softmax. So, we proposed to normalize it with some probability density function (PDF) scale, such that $\int PDF = 1$. However, if used straight-forwardly, it would need to involve a division operation, what is strongly un-desirable for devices with constrained computational power. Therefore, instead of dividing, we propose to substitute division by multiplication with some PDF normalizing constant as below:

$$\sigma(x_i) = \frac{\sigma^*(x_i)}{PDFnorm} \to \sigma^*(x_i) \cdot \alpha \tag{5}$$

where $\alpha = e^{-ln(\Sigma\sigma^*(x_i))}$ is PDF normalizing constant.

Then final equation to compute softmax approximation by proposed REXP method is shown below:

$$\sigma(x_i) = \frac{e^{-ln(\Sigma\sigma^*(x_i))}}{e^{max(x)-x_i}} = \frac{1}{e^{max(x)-x_i}} \cdot e^{-ln(\Sigma\sigma^*(x_i))} \tag{6}$$

Thus, to compute the softmax value it requires just two LUTs of considerably small size, where content of the first LUT is computed accordingly to Eq.(4), and the second LUT values can be computed as below:

$$LUT_\alpha[j] = \left\lfloor \frac{1}{j} \cdot (2^w - 1) \right\rfloor, \forall j = 0, 1, ..., x_s - 1 \tag{7}$$

where $j = \Sigma\sigma^*(x_i)$, $x_s$ is selected quantization boundary, and $LUT_\alpha[x_s] = 0$.

## 4.2   2-Dimensional LUT

In this subsection we propose another method which is based on the substitution of a division operation in Eq.( 2) by 2-Dimensional (2D) LUT to speed-up and simplify the computation, while maintaining accuracy even for attention-based DNN models. Hereafter we will refer to this method as 2D LUT.

For this purpose, we have started with the estimation of distributions of $e^x$ and $\Sigma e^x$ terms for typical inference runs. Our investigation showed that if max-based normalization is applied to the input values (i.e., $x \to (x - \max(x))$), the distribution of $e^x$ is stable within range $e^x \in (0, 1]$ regardless of the input values, and range of $\Sigma e^x$ term depends on the length of the input $x$. Thus, it allows us to have stable computation even within small size of LUT.

Generic architecture and concept of the proposed method for efficient softmax implementation as 2D LUT is shown in Figure 1. There are two LUTs used: 1D LUT for approximation of $e^x$ values, and 2D LUT for storing softmax output values dependent on the values of numerator $e^x$ (used as the 1-st index in the table), and denominator $\Sigma e^x$ (used as the 2-nd index) of Eq.(2). As it can be seen from Figure 1(right), to calculate the indexes for corresponded value in 2D LUT table, only most-significant bits (MSB) are needed. Thus, the simplest hardware realization can be done within wiring only (when MSB bits are directly connected to the appropriate address selectors) [3]. Also, the proposed method can be easily modified to the case where, 1-st index of 2D LUT table is calculated not from $e^x$ but directly from input $x$. In such case there is no need to store intermediate values of $e^x$.

While the content of 1D LUT for approximation of $e^x$ values is straightforward, 2D LUT contains the family of linear approximations where each row contains the softmax output scaled according to

---
[3]Other hardware realization are also possible, but not considered here for simplicity of the explanation.

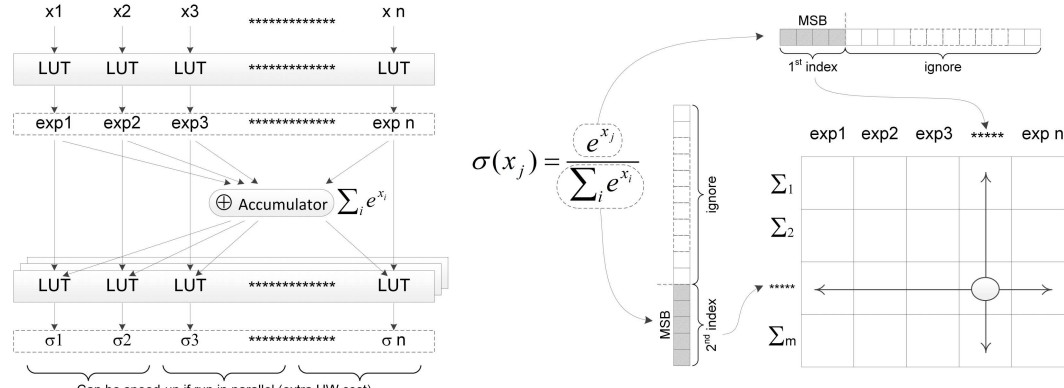

Figure 1: Generic concept of the proposed 2D LUT method (left). Reading softmax output value from pre-computed 2D LUT(right). Computational flow consists from two steps: a) obtaining of $e_i^x$ values by reading from 1D LUT and accumulation of $\Sigma e^x$ term, b) obtaining $\sigma(x_i)$ values by reading from 2D LUT.

$\Sigma e^x$ term as shown in Eq.(8). The indexes of LUT are computed according to Eq.(9) and Eq.(10), $w$ means the number of bits for the value in selected precision.

$$LUT_\sigma[i][j] = \left\lfloor \frac{i \cdot scale_{e^x}}{j \cdot scale_\Sigma} \cdot (2^w - 1) \right\rceil \tag{8}$$

where

$$i = 0, ..., \left\lfloor \frac{max(e^x)}{scale_{e^x}} \right\rceil \tag{9}$$

$$j = 1, ..., \left\lfloor \frac{max(\Sigma e^x)}{scale_\Sigma} \right\rceil \tag{10}$$

Since $x \to (x - \max(x))$ normalization was used, so $max(e^x) = 1.0$. Therefore, $scale_{e^x}$ factor allows to define the number of columns in LUT to make it small enough for practical applications. In our experiment we have selected $scale_{e^x} = 0.1$ for all precisions, what allows us to reduce the size of LUT significantly (i.e., $i = 0, ..., 10$ for all versions of $LUT_\sigma$). The value of $max(\Sigma e^x)$ depends on the distribution of input values. Our experiments showed that $max(\Sigma e^x) = 60$ is big enough for the tested NLP applications. We also selected $scale_\Sigma = 1.0$ for simplicity of the computations. Thus, finally, those parameters give us $LUT_\sigma$ of typical size $11 \times 60$.

## 5   Experimental validation

To validate the proposed methods and check how well they generalize we have conducted several experiments with different models (DETR, Transformer, and BERT) for different applications (object detection, machine translation, sentiment analysis, and semantic equivalence) over variety of datasets. In all those experiments we have used available pre-trained models, where we applied dynamic post-training quantization (hereafter we referred to quantized models as PTQ-D) [4]. Then we substituted a conventional softmax layer in quantized models with the LUT-based computation as described in Section 4. We did not consider any retraining or fine-tuning of the models after quantization, and the same off-line generated LUTs were used among all models. Our code allows to select LUTs with different precision from int16 down to uint2, what allows to analyze the sensitivity of the model to softmax approximation even for ultra-low 2-bits quantization. The details of experiments are described below, and results are summarized in Figure 2, Figure 3, and Table 2 . For more details

---
[4]See more details in Appendix A.3.

please refer to Table 6, and Table 7 in Appendix. As it can be seen from figures, proposed LUT-based softmax computation methods maintain accuracy drop below 1.0% down to 8-bit quantization for all NLP and DETR (no DC5) models.

## 5.1 Object detection

For our first experiments we have used DEtection TRansofmer (DETR) models for object detection [2], with available pre-trained models [5]. As it can be seen from Table 6, we were able to reproduce the same results for original FP32 reference model over COCO dataset. We have used the same IoU metric by Average Precision (AP) as in Table 1 in [2]. Then we run a bunch of experiments to check how accuracy of object detection will be decreased due to PTQ-D quantization and LUT-based approximation as proposed in REXP method (see Section 4.1). Table 5 in Appendix shows the LUTs size for several pre-selected cases in int16 and uint8 precision. There are three cases selected which are different in the size of $LUT_\alpha$: it is $1 \times 256$ for case 1, $1 \times 320$ for case 2, and $1 \times 512$ for case 3.

Analysis of Figure 2 shows that accuracy drop caused by application of softmax approximation is small ($< 1\%$) and acceptable for plain DETR models (no DC5 used). Bigger accuracy drop for +DC5 cases is caused by the bigger size of self-attentions of the encoder (see details in Section 5.3). We expect that increasing size of LUTs will help to solve this issue. The behavior of average recall values is similar to average precision values.

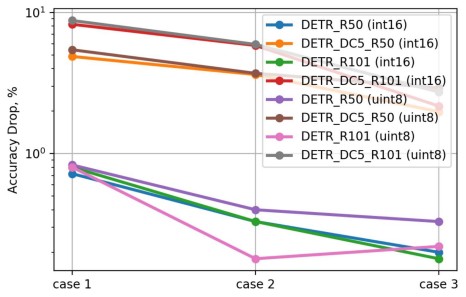 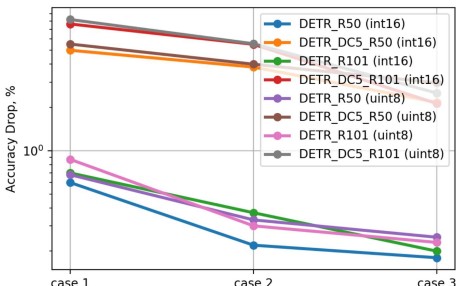

Figure 2: DETR averaged accuracy drop of PTQ-D models with softmax approximations vs. original FP32 models: average precision (left) and average recall (right). As it can be seen from the figure, for DETR models without dilation at the last stage (no DC5) the accuracy drop for all cases is below $1\%$ and shows very similar behavior.

## 5.2 NLP tasks

Next, we have validated the proposed methods by experimenting with several NLP tasks. Similarly, to DETR case, Table 8 in Appendix shows the LUTs size for several pre-selected cases for those experiments. In Table 2 below there are accumulated values from the experiments with NLP models. The bold values in the table shows highest values per model per method after applying quantization and softmax approximation. As it follows from the analysis of experiment results, about 700 Bytes for 2D LUT method, and up to 50 Bytes for REXP method would be enough for practical applications.

### 5.2.1 Machine Translation

Among NLP tasks we have started with machine translation. For our experiments we have used transofmer-base model for En-Ge translation [12] from OpenNMT library, with available pre-trained model [6], configured to replicate the results from original paper. To avoid dependency of the evaluation results on the selected tokenization scheme, we have used `spm_decode` [7] to detokenize the output of translation, and then applied `multi − bleu.perl` script [8] to calculate BLEU score.

---

[5]https://github.com/facebookresearch/detr

[6]https://opennmt.net/Models-py/

[7]https://github.com/google/sentencepiece

[8]https://github.com/moses-smt/mosesdecoder

Table 2: Experimental validation over different NLP models and datasets

| PRECISION | TRANSFORMER | | | | BERT | | | |
|---|---|---|---|---|---|---|---|---|
| | 2D LUT | | REXP | | 2D LUT | | REXP | |
| | WMT 2014 (BLEU) | WMT 2017 (BLEU) | WMT 2014 (BLEU) | WMT 2017 (BLEU) | SST-2 (%) | MRPC (F1) | SST-2 (%) | MRPC (F1) |
| FP32 | 26.98 | 28.09 | 26.98 | 28.09 | 92.32 | 90.19 | 92.32 | 90.19 |
| PTQ-D | 26.86 | 27.95 | 26.86 | 27.95 | 91.74 | 89.53 | 91.74 | 89.53 |
| INT16 | **26.87** | **28.02** | **26.89** | 27.64 | **91.63** | **89.50** | **91.74** | 89.26 |
| UINT8 | 26.76 | 27.9 | 26.8 | **27.66** | **91.63** | 89.35 | 91.17 | **89.34** |
| UINT4 | 26.26 | 27.43 | 26.68 | 28.02 | 91.40 | 88.01 | 91.17 | 88.77 |
| UINT2 | 24.42 | 25.06 | 25.29 | 25.86 | 89.22 | 56.67 | 91.63 | 86.12 |

Thus, as it can be seen from Table 2, we were able to reproduce the same BLEU score for FP32 reference model as in original model. Then we run several experiments to check how accuracy of the translation will be changed due to LUT-based quantization in different precisions, and we can confirm that down up to 8-bit quantization the deviation of BLEU score from reference is small for both datasets ($< 0.5\%$). Also, if we consider impact of the proposed methods only, then we can see that accuracy drop is much smaller, and sometimes even recovers vs. PTQ-D quantization (see Figure 3 (right)).

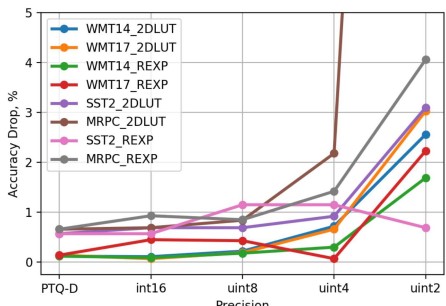 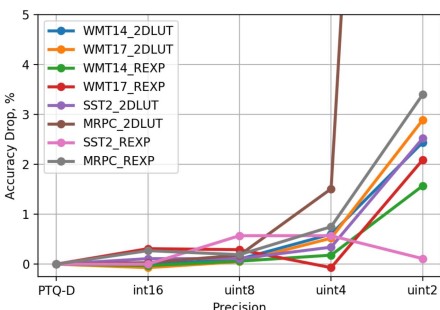

Figure 3: Accuracy drop for NLP experiments: PTQ-D models + softmax approximations vs. FP32 models (left), and PTQ-D models + softmax approximations vs. plain PTQ-D models (right). As it can be seen from the figure, down to uint8 precision the accuracy drop for all cases is below $1\%$ and shows very similar behavior. This confirm very good generalization of the proposed method over different models and applications.

### 5.2.2 Sentiment analysis

To extend the variety of NLP applications, we also tested the same LUTs with BERT model [5]. We have used sentiment analysis task from GLUE benchmark [29] to test the model. We have used `huggingface` library [9], and trained the model with the hyper-parameters described in [10]. The results of our experiments showed, that similarly to machine translation, the impact of proposed method (softmax layer approximation by LUTs) is smaller vs. accuracy drop caused by PTQ-D quantization (see Figure 3).

### 5.2.3 Semantic equivalence

For semantic equivalence test we used The Microsoft Research Paraphrase Corpus (MRPC) [11] in GLUE benchmark. As the classes are imbalanced (68% positive, 32% negative), we follow the

---

[9]https://github.com/huggingface/transformers
[10]https://github.com/google-research/bert
[11]https://www.microsoft.com/en-us/download/details.aspx?id=52398

common practice and used F1 score as a metric. We have used `huggingface` library and followed the guidelines from PyTorch tutorial [12] to obtain PTQ-D quantized model. Then, similarly to previous tests we have substituted a conventional softmax layer with the proposed LUT-based methods. The results of our experiments showed the similar trend with sentiment analysis test.

## 5.3 Ablation study of DETR models experiment

As it is stated in [2], to increase the feature resolution for small objects, a dilation to the last stage of the backbone was added (+DC5 cases of DETR models). This modification increases the cost in the self-attentions of the encoder, leading to an overall $\times 2$ increase in computational cost. Such changes also reflect on the properties of softmax factors. In Figure 4 there are shown the histogram of $\Sigma e^x$ values distributions for the first 200 tensors of DETR model run for `bins = 50, range = (0, 500)`. As it can be seen from the figure, the distribution of DETR+DC5 (R50) variant is more right-tailed, due to the bigger number of high-magnitude values. This causes the bigger accuracy drop when LUT-based quantization method is used, due to the lack of the discrepancy for those values. Thus, for such models (DETR with added dilation at the last stage) the accuracy of object detection after application of the proposed method can be limited. However, as we can see from Figure 2) increasing of the size of $LUT_\alpha$ from 256 Bytes to 512 Bytes allows to decrease the accuracy drop from 9% to 3% for DETR+DC5 (R101) unit8 case. Thus, we expect that further increasing the size of LUTs will help to obtain even more accurate results for DETR models with dilation.

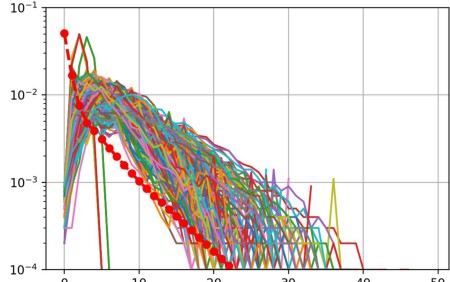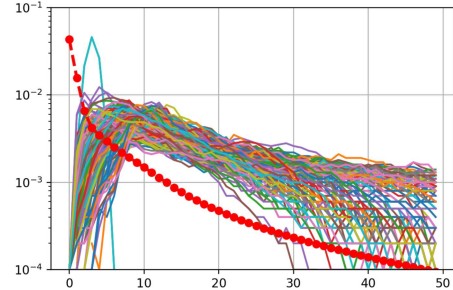

Figure 4: Histogram of $\Sigma e^x$ values distributions for DETR model variants: plain DETR (R50) (left), and with dilation DETR+DC5 (R50) (right). Red dot line represents the average of the all values per one run of the inference of DETR model. It is clearly seen from the figure that distribution of DETR+DC5 (R50) variant is more flat, having more high-magnitude values. This causes the bigger accuracy drop for the quantized model due to lack of the discrepancy for those values.

## 6 Conclusion

In this paper two alternative methods for efficient softmax computation for DNN models with attention mechanism are proposed. The methods are memory-centric in contrast to known logic-centric approach and are based on the usage of LUTs for reading of the pre-computed values, instead of the direct computation. Thus, it allows to build the HW accelerator without usage of costly and power-hungry divider. In turn, it allows to decrease the power consumption and latency of the whole inference, what is crucial for edge computing. All results obtained in the paper were validated over different AI tasks (object detection, machine translation, sentiment analysis, and semantic equivalence) and models (DETR, Transformer, BERT) by variety of benchmarks (COCO2017, WMT14, WMT17, GLUE), showing acceptable accuracy and good generalization of the proposed methods.

---

[12]https://pytorch.org/tutorials/intermediate/dynamic_quantization_bert_tutorial.html

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
