# OpenReview forum: "Efficient SoftMax Approximation for Deep Neural Networks with Attention Mechanism"
_NeurIPS.cc/2021/Conference — NeurIPS 2021 Submitted_

### Official Review · Reviewer_3inJ · 2021-06-26

**Rating:** 4
**Confidence:** 4

**Summary:**

This paper presents two hardware-friendly (no floating points or divisions) quantization methods for softmax computation. It is particularly targeted at the use in attention layers in transformer models, where the output precision of the softmax layer is critical to the overall accuracy of the model.

**Limitations And Societal Impact:**

Yes, the authors have clearly measured the accuracy drop incurred by the use of their method, particularly when using dilation layers in the DETR model.

**Main Review:**

Note that I was one of the reviewers of a previous version of this paper submitted to ICML 2020.

It is appreciated that the new experiments are done on fully quantized models, which makes more sense than having only the softmax quantized while the rest is computed in FP32.
Indeed a fully quantized model could run on an FPGA for example, where not only divisions are to be avoided, but also floating point operations in general.

While the LUT-based methods presented are not extremely revolutionary, the experimental results prove that they are viable for use in complex transformer models while incurring an accuracy drop of about a percent, which could be acceptable in certain edge compute scenarios.

However the paper is hard to follow, notably because of the numerous typos and grammatical errors. It is particularly alarming considering that the errors I sampled in my ICML 2020 review, for which I provided corrections, **are still present**:
- (line 28) "up to 13X times" -> "up to 13 times" -- as "X" stands for "times".
- (line 68) "and sequence length is" -> "and the sequence length is"
- (eq 2) should be $ \sigma (x_j) = \frac{e^{x_j}}{\sum_{i} e^{x_i}} = \frac{e^{x_j - max(x)}}{\sum_{i} e^{x_i - max(x)}} $
- (line 197) "to calculate the indexes for corresponded values in 2D LUT table" -> "to calculate the indices for the corresponding values in the 2D LUT" -- other than the 2 grammatical values, note that LUT stands for Lookup Table.
- (line 199) "MSB bits" -> "MSBs" -- MSB stands for Most Significant Bit.
- (line 211) "60 is big enough" -> "60 is large enough" -- bad choice of word.
- (line 233) "we run a bunch of experiments" -> "we run multiple experiments" -- bad choice of word.
- Anywhere "what" is used but not in a question, should use "which" instead.
- The bibliography entries are using different formats, and some are missing information.

While 2 solutions are presented, 2D-LUT and REXP, it is not clear if the goal of the paper is to compare them. Indeed the NLP experiments do compare the performance of the two solutions, however the object detection experiments using DETR seem to rely only on REXP. Is there a particular reason for this? Is the 2D-LUT somehow problematic in the DETR case?

Other presentation issues:
- The plots in figure 2 and 3 are not easy to read. And more generally, it is preferred to use alternatives to color-coding, since it is common to print in black and white, and could also pose problems for color-vision impaired persons.
- In plot 4, the averages would have been good enough. Having tens of superimposed lines is not helping with interpreting the data.


**Time Spent Reviewing:**

3

---

> ### Author Response · Authors · 2021-08-10
> **Thank you for the valuable comments on ICML 2020 submission. Mostly due to those comments we heavily re-wrote the paper. To improve grammar we will use an external editor before submission of the final paper.**
>
> We gratefully appreciate and thank anonymous reviewer 3inJ for pointing out the parts of our paper, where lack of the clarity complicates the understanding. We also want to thank one more time for his/her valuable comments on our earlier ICML 2020 submission. Mostly due to those comments we heavily re-wrote the paper, added additional experiments (DETR) and make all of our experiments on quantized (PTQ-D) models, as it was suggested.
>
> 1. While 2 solutions are presented, 2D-LUT and REXP, it is not clear if the goal of the paper is to compare them.
>
> In our paper we have proposed two alternative methods (line 293), and final choice depends on the HW platform available. If there is enough memory (typical case for DRAM-based accelerator, where a huge amount of memory cells already exists and can be used for storing the corresponding values), -- then 2D LUT method can be used, potentially avoiding not only a divider but even a multiplier, further reducing the latency and area. In another case If utilization of HW multiplier is preferable, -- then REXP method can be used (in this case smaller amount of memory for LUTs is required).
>
> 2. Indeed the NLP experiments do compare the performance of the two solutions, however the object detection experiments using DETR seem to rely only on REXP. Is there a particular reason for this? Is the 2D-LUT somehow problematic in the DETR case?
>
> There is not any fundamental problem to use 2D-LUT method for DETR case. Naturally, 2D LUT method requires bigger size of LUTs for the implementation (e.g., see Table 8 to compare the typical size of LUTs for 2D LUT and REXP methods in NLP experiments). Therefore, we limited our experiments for REXP method only.
>
> 3. The bibliography entries are using different formats, and some are missing information.
>
> For all bibliography we have used BiBtex entries (generated from arXiv) and use Latex style to automatically generate and append it to the paper.
>
> 4. The other comments are related to grammar, figures,  improper choice of words, etc.
> We will use an external editor before submission of the final paper.

---

### Official Review · Reviewer_BDef · 2021-07-09

**Rating:** 4
**Confidence:** 5

**Summary:**

The authors propose an efficient computation of softmax targeting transformer models running on a specialized hardware accelerator. Softmax is a key component of the self-attention mechanism of transformers and must be applied to a sequence length x sequence length sized attention score matrix. It can be a key bottleneck for hardware accelerators.

The proposed idea uses a 2D look-up table (LUT) indexed by most significant bits in the numerator and denominator of softmax. This avoids using a divider and allows the inputs to be in integer precision. When using 16 bits of precision, there is little to drop in accuracy on a variety of transformer NLP tasks with the approximate softmax.


**Limitations And Societal Impact:**

No negative social impact.

**Main Review:**

I have two main issues with the paper:

First, I don't get the point of most of the derivations in Section 4.1. Take Equation (3):
$$
\sigma^*(x_i) = \frac{1}{e^{max(x)-x_i}}
$$
This formula is the same as the softmax numerator:
$$
\sigma^*(x_i) = \frac{1}{e^{max(x)-x_i}} = e^{x_i-max(x)}
$$

Now take Equation (6):
$$
\sigma(x_i) = \sigma^*(x_i) * e^{-ln(\sum{\sigma^*(x_i)})}
$$
The e^-ln(x) part is just 1/x, since \sigma^*(x_i) is always positive. So Equation (6) reduces to
$$
\sigma(x_i) = \frac{\sigma^*(x_i)}{\sum{\sigma^*(x_i)}}
$$
But that's exactly the same as the regular softmax formula.

I'm not sure what the point of the whole section is. From what I understand, LUTs are used to approximate e^(-x) and 1/x, see Equations (4) and (7). But the core softmax algorithm is unchanged. If this is correct, section 4.1 should be rewritten to be more clear, it is currently very confusing. The section begins by discussing REXP which seems irrelevant (1/e^x = e^(-x)). Once the notation is understood, the technique is basically to approximate the expensive ops in softmax (exponential and division) with LUTs.

Second, there is no hardware comparison or any sort of metric, despite this being a technique that only works for specialized HW accelerators. The idea of a 2D LUT sounds promising, but the LUT would have W^2 entries (W being the bitwidth), which is a lot. The LUT will achieve better throughput and latency than a HW divider, but how about the area cost? Dividers are necessary for other operations in a Transformer (e.g., layernorm) so would the LUT be replacing dividers or costing additional area? There are so many tradeoffs to consider and experiments to do in HW before an idea like this could be considered promising. This paper does not mention these points.

This being NeuRIPS, I don't expect an RTL hardware evaluation. But the paper claims that 2D LUTs are better than dividers - I need  to see some kind of hardware comparison between them. Area, throughput, energy, etc. The most novel part of the paper is the HW architecture (replace divider with LUT), but there is no architecture evaluation to convince readers.

**Time Spent Reviewing:**

2

---

> ### Author Response · Authors · 2021-08-10
> **We intentionally made our Eq.(6) to be mathematically equivalent to the original softmax formula, but we have changed the computational flow to avoid a division operation.**
>
> We gratefully appreciate and thank anonymous reviewer BDef for pointing out the parts of our paper, where lack of the clarity complicates the understanding.
>
> 1. First, I don't get the point of most of the derivations in Section 4.1. Proposed formula is exactly the same as the regular softmax formula.
>
> Yes, you are correct, -- our Eq.(6) is mathematically equivalent to the original softmax formula, and it is our intention. In our paper we do not change the softmax function, but instead we propose a special way to compute it intentionally avoiding dividers, which are very expensive in the Edge/IoT computing devices. Therefore, we have changed the computational flow to eliminate division operation, and thus exclude (typically heavy) divider from HW design of Accelerator for Edge/IoT NPU. Also, substitution of e^(-x) operation by mathematically equivalent 1/e^x operation allows to apply LUT-based method and use ‘x’ directly as an index in the LUT.
>
> 2. Second, there is no hardware comparison or any sort of metric. The idea of a 2D LUT sounds promising, but the LUT would have W^2 entries, which is a lot.
>
> The required sizes of LUTs for the proposed methods are represented in Table 5 and Table 8, and it is shown that around 700 Bytes of memory cells are enough (Please also note, that for NLP tasks for REXP method even 24 Bytes are enough). Moreover, for DRAM-based accelerator LUT costs are almost free (see lines 144-145), since there is already a huge amount of memory cells in DRAM which can be used for storing the corresponding values.
>
> 3. Dividers are necessary for other operations in a Transformer (e.g., layernorm) so would the LUT be replacing dividers or costing additional area?
>
> We are applying our memory-centric idea to Layer normalization as well. It is in the progress, and results are also promising.
>
> 4. This being NeurIPS, I don't expect an RTL hardware evaluation. But the paper claims that 2D LUTs are better than dividers - I need to see some kind of hardware comparison between them.
>
> Yes, as our paper is for NeurIPS conference and we avoid using HW-specific information as inappropriate. Our main hardware-like comparison is based on the required size of LUTs. And as we just replied above it is small and can be almost free for DRAM-based accelerators.

---

### Official Review · Reviewer_GRd4 · 2021-07-18

**Rating:** 7
**Confidence:** 4

**Summary:**

Current paper raises the necessity of efficient softmax approximation for custom hardware for attention-based models. Currently softmax computation requires data move to host and then back which is expensive operation. Existing methods on softmax approximation are applicable in case when softmax is at the end of computational graph. First, current paper demonstrates that existing methods on softmax approximation are not applicable to attention-based methods (performance degradation is huge or even a model does not work anymore, e.g. DETR gives 0 mAP). Second, authors propose two approaches on efficient softmax approximation, which is memory-centric: one is with two look-up tables (LUT) and multiplication and another one is with 2d LUT and no multiplication/division. Size of LUT is controllable parameter and depends on the size of attention in a model to have enough capacity to support different values. It is shown that for variety of tasks (object detection, machine translation, sentiment analysis, semantic equivalence) and models (DETR, BERT, Transformer) 8-bit approximation allows to obtain acceptable accuracy loss below 1.0%. Experimentally it is analyzed that ranges of numerator and denominator of softmax are stable if max normalization is applied to the input and LUT table with quite small size is enough (700 Bytes).

**Limitations And Societal Impact:**

Authors discussed limitations of their work throughout the paper (that proposed approach is HW hardware oriented) and in section 5.3 when encoders with bigger self-attentions are used what causes larger variation in values of $\sum e^x$ (used LUT sizes cover small range of different values).

There is no potential negative societal impact as the paper focuses on inference optimization.

**Main Review:**

**Originality**

There are a lot of works on softmax approximation (covered well in the paper), mainly to use max normalization and logarithmic reformulation with further look-up tables (LUT) usage. Current paper also follows widely used technique on LUT, however via existing reciprocal exponentiation method authors approximate the softmax in a novel way. Before proposing a new methodology authors perform in-depth analysis of existing methods on softmax approximation and demonstrate that even logarithmic reformulation with max normalization is not applicable for attention-based models (performance drop is huge).

**Quality**

Report on prior art benchmark looks reasonable. Math on how softmax approximation is done is correct, however some suggestions on readability and typos are listed below. Proposed approximation benchmarking looks reasonable having in mind that it is performed and compared after model quantization which is a realistic scenario. Inference benchmark covers several models and domains, especially heavy DETR model, so in that respect experiments are convincing enough. Observed large performance degradation for DETR with bigger size of self-attentions is discussed in section 5.3: larger LUT sizes are not tested, however the trend is seen in Fig.2 and already 512 Bytes LUT provides <3% performance degradation.

**Clarity**

Paper mostly well written, however some parts/plots could be simplified (see suggestions below in the comments section) to improve readability. The paper focuses on the inference. All information about models used for benchmark (links on pre-trained open-sourced models, code-base to run them and quantization tutorial from pytorch) and implementation details on proposed algorithms + LUT structure and precision manipulations are given in the paper and Appendix.

**Significance**

Significance of the work is in-depth discussed in sections 2 and 3 in the paper. While GPU/TPU already supports softmax, HW accelerators for the inference at the edge/IoT devices with limited computational power do not support it and the softmax becomes a bottleneck for attention-based models where it appears not only at the end of computational graph but in all intermediate attention-blocks. The paper proposes two ways on stable softmax approximation for the cases when softmax is extensively used in the intermediate operations (not only at the end) so that HW will have no extra time consuming operations on moving data to CPU host and then back for softmax computation. This opens efficient deployment of attention-based models on HW hardware, for example on mobile. Proposed approximation allows to obtain acceptable accuracy loss below 1.0% for the variety of tasks and models.

Comments:
- why is speech recognition mentioned in the abstract while there is no any tests for speech recognition models in the paper?
- line 66: how did authors compute the number of softmax operations as N*L*L? More precise is to say that there are N*L softmax operations where for each softmax operation we need to perform L exp operations and L divisions. Otherwise the original statement in the paper is misleading. Or formulate that O(N*L*L) arithmetic operations are needed.
- line 80 Eq.2: The correct way to write down is to put $j$ index in the denominator in the summation
- I think lines 171-178 and eq. (5)-(6) are redundant and does not make a clear point in the text. Authors could simply write down that let's represent eq. (2) as multiplication operation: $\sigma(x_i) = \sigma^*(x_i) \cdot \frac{1}{\sum\sigma^*(x_k)}$, where first multiplier can be computed with LUT eq.(4) and the second multiplier can be computed by another LUT defined by eq.(7)
- Be consistent on usage index $i$/$j$ in the text/figures for numerator/denominator of eq. (2)
- Section 4.2 can be also simplified. 188-192 lines is better to move into sec. 4.1 as this provides intuition why 2 LUT tables approach (sec 4.1) works (because of stability of distributions). Then 193-201 can be simplified because it is clear to say "let's save now even multiplication operation in eq.(6) by pre-computing 2D LUT as LUT1 $\times$ LUT2. And now as indices we even can use directly x and $\sum e^x$".
- "indexes" -> "indices"
- Suggestion: legends in Figure 2 could be put outside the plot, int16 can be solid, while int8 can be dashed  but with the same color or mark DC5 with the same color as no_DC5 but dashed.
- For Fig. 2 would be good to add plot for PTQ-D variant as it is shown in Figure 3.
- Could authors explain why in Table 2 2D LUT for BERT uint2 F1 is so bad compared to REXP? Do authors have any intuition when LUT 2D is better than REXP in terms of accuracy (if we assume that for hardware both have the same real time)?
- For Fig. 3 for readability make 2DLUT models solid and REXP make dashed and the same color as correspondent 2DLUT.
- I have checked https://opennmt.net/Models-py/ and WMT 14 has 26.89 BLEU, while in Table 2 it is reported 26.98. Is it typo or reevaluation artifact?
- It is necessary to include results on Semantic equivalence too (at least in the Appendix).
- line 282: "there are shown the histogram of ..." -> "the histogram of ... is shown"
- line 472: "In Table 3 there are shown results of experiments" -> e.g. "Table 3 shows results of experiments"
- line 477: "Eg.( 12)" -> "Eg.(12)"
- line 483: "These codes mimics" -> "These codes mimic" or "This code mimics"
- Algorithm 1, line 9-11: why do we need $idx_{e^{x_i}}$? why the line 11 is not like $\sigma(x_i)=LUT_{1/e}[idx_{x_i}]\cdot LUT_\alpha[idx_\alpha]$
- what about real-time testing on HW hardware? (I believe this is not really necessary, but curious to see how not approximated softmax slows down the inference in practice).

**Time Spent Reviewing:**

6

---

> ### Author Response · Authors · 2021-08-10
> **Thank you for the detailed review and excellent comments**
>
> We deeply appreciate and sincerely thank anonymous reviewer GRd4 for so detailed review of our paper (including Appendix section) with the excellent comments which encourage us and help to improve the quality of the paper.
>
> 1. Why is speech recognition mentioned in the abstract?
>
> It is just remained typo from previous version of our paper submission. Since quantized speech recognition model is proprietary, we had to remove anything related to speech recognition task.
>
> 2. Could authors explain why in Table 2 2D LUT for BERT uint2 F1 is so bad compared to REXP?
>
> 2-bit quantization is just too harsh for the accuracy of the model and big deviation between the results of the different methods is natural. At this moment we have no clear answer and we plan more tests including 3-bit quantization.
>
> 3. I have checked https://opennmt.net/Models-py/ and WMT 14 has 26.89 BLEU, while in Table 2 it is reported 26.98. Is it typo or reevaluation artifact?
>
> Yes, it is the re-evaluation artifact (we have retested it several times at our side to re-confirm it).
>
> 4. It is necessary to include results on Semantic equivalence too (at least in the Appendix).
>
> In Table 2 we represented Semantic equivalence results as "MRPC". Probably direct referring to it as "see Table 2" would be clearer.
>
> 5. Algorithm 1, line 9-11: why do we need idx_e^xi? Why the line 11 is not like σ(xi)=LUT1/e[idxxi]⋅LUTα[idxα]
>
> Yes, your feedback is helpful, and line 11 can be simplified as you have proposed. Since Algorithms 1 and 2 are just pseudocode to represent the computational logic, we thought that using idx_e^xi would be more natural and less confusing to a reader.
>
> 6. What about real-time testing on HW hardware?
>
> Since design of our HW accelerator is proprietary, at this moment none of such information can be revealed.
>
> We agree to and will reflect all of your other comments and advices related to grammar, figures, and clarity in the final paper.
>
> Thank you again.

---

### Official Review · Reviewer_4L69 · 2021-07-21

**Rating:** 5
**Confidence:** 4

**Summary:**

As the self-attention models have been widely used in NLP and computer vision community, a cost efficient implementation of softmax is becoming critical. This paper proposes two methods based on LookUp Table (LUT) to approximate the softmax computation. The proposed method shows generalise ability on a number of tasks including object detection, machine translation, speech recognition and semantic equivalence.

**Limitations And Societal Impact:**

see above

**Main Review:**

My most concern is that the paper lacks of baseline methods to validate its superiority. It only compares with easy baselines .

Big accuracy drop occurs when large self-attention matrix is needed but the paper only gives hypothesis rather than experiments -- "we expect that increasing size of LUTs will help to solve this issue" (L241).

**Time Spent Reviewing:**

4

---

> ### Author Response · Authors · 2021-08-10
> **Since our focus is an inference at Edge/IoT devices, we have selected those baseline methods, which are not using a division operation.**
>
> We gratefully appreciate and thank anonymous reviewer 4L69 for pointing out the parts of our paper, where lack of the clarity complicates the understanding.
> 1. Lack of baseline methods to validate superiority:
>
> As it is stated in lines 128-130 the focus of our paper is HW accelerators for the inference “… at the edge devices, which usually have limited computational power …” and additionally in lines 452-452 we have stressed that “we are interested in the methods which are not using a division operation”. Thus, for the baseline we have considered only those prior arts where logarithmic transformation was applied. Therefore, we have conducted a number of experiments which are summarized in Table 1. However, more details of the prior arts experiments can be seen in Appendix A.1.1 and A.1.2 , including Table 3.
>
> 2. Big accuracy drop occurs when large self-attention matrix is needed but the paper only gives hypothesis rather than experiments:
>
> This case is related to 5.3 Ablation Study section, from where we have found out that the bigger accuracy drop is caused by application of dilation to the last stage of the backbone of DETR model. As it is stated in lines 288-290 “increasing of the size of LUT from 256 Bytes to 512 Bytes allows to decrease the accuracy drop from 9% to 3% for DETR+DC5 (R101) unit8 case”. Thus, we naturally assume that further increasing the size of LUTs will reduce accuracy drop even more. We have not showed those experiments because of the space limitation of the paper.
> Please also note, that for DETR models without dilation at the last stage, the accuracy drop is still below 1% (see Figure 2 and Table 6).

---

### Decision · Program_Chairs · 2021-09-27

**Decision:**

Reject

**Comment:**

The paper presents two efficient 8 bits approximations for the softmax operation in the context of self-attention, and benchmarks it in object detection, machine translation, speech recognition, semantic equivalence. The loss of accuracy is minimal, the downstream task effects evaluation is adequate. The value of the contribution is focused on custom hardware. There is a thorough review (GRd4) that is supportive of acceptance, but the other reviewers were not convinced. In particular, the clarity of the paper, the hardware evaluation, as well as contextualization, can be improved further.

Overall, the contributions look interesting and promising, but the paper is not in a state ready for publication at NeurIPS.